# Exploration of Microalgae-Activated Sludge Growth Performance in Lab-Scale Photobioreactors under Outdoor Environmental Conditions for Wastewater Biotreatment

Abraham O. James [1,2], Abayomi O. Bankole [1,3], Caroline M. E. Pompei [1], Gustavo A. S. A. Dantas [1], Graziele Ruas [1,*] and Gustavo H. R. Silva [1]

1   Department of Civil and Environmental Engineering, School of Engineering, São Paulo State University (UNESP), Bauru Campus, Sao Paulo 17033-360, Brazil; abraham.o.james@unesp.br (A.O.J.); abayomi.o.bankole@unesp.br (A.O.B.); caroline.erba-pompei@unesp.br (C.M.E.P.); gustavo.alencar@unesp.br (G.A.S.A.D.); gustavo.ribeiro@unesp.br (G.H.R.S.)
2   Department of Environmental Management and Toxicology, Federal University of Agriculture, Abeokuta 111101, Nigeria
3   Department of Water Resources Management and Agrometeorology, Federal University of Agriculture, Abeokuta 111101, Nigeria
*   Correspondence: graziele.ruas@unesp.br

**Abstract:** Increasing the volume of untreated and inadequately treated municipal wastewater undermines the circular economy potential of wastewater resources, particularly in low-income regions. This present study focused on and evaluated the performance of native microalgae-activated sludge (MAS) growth for tertiary treatment of anaerobically digested wastewater from an up-flow anaerobic sludge blanket (UASB) in an outdoor lab-scale photobioreactor (2.2 L). Three conditions with distinct MAS inoculum concentrations alongside three controls were operated in batch mode for 5 days hydraulic retention time (HRT) at 11.5:12.5 photo-hours. The MAS inoculum concentration influenced the treatment outcome. The best performance was observed when the MAS concentration was 0.10/0.20 g L$^{-1}$, and the cell density was $1.60 \times 10^7$ cells mL$^{-1}$, total biomass productivity of 0.10 g TSS L$^{-1}$ d$^{-1}$, total phosphorus uptake of 85.1%, and total nitrogen uptake of 66.1%. Logarithmic removal (Log-Re) of bacterial pathogens (water quality indicators) showed Log-Re 3.4 for total coliforms ($1.37 \times 10^2$ CFU 100 mL$^{-1}$) and 4.7 for *Escherichia coli* ($0.00 \times 10^0$ CFU 100 mL$^{-1}$). The results revealed optimum remediation performance and nutrient recovery potential with appropriate inoculum concentration, in admiration to advancing the science of circular economy.

**Keywords:** microalgae-activated sludge; nutrients recovery; pollutant removal; total coliform; *Escherichia coli* removal





## 1. Introduction

Currently, approximately 80% of wastewater is released into the environment globally without adequate treatment [1], which poses a major challenge confronting the whole world. The volumetric increase in untreated wastewater released into the environment from municipal, industrial, and agricultural activities may be exacerbated due to the unmatched capacities of extant treatment facilities, mediated by the human population explosion and increased consumption from continuously changing lifestyles [2]. This constitutes a significant threat to the quality of water resources and public health [1–6], and by implication may affect the attainment of the United Nations' Sustainable Development Goals (SDGs).

The use of microalgae-activated sludge (MAS) bacteria co-culture has been shown to be a promising alternative wastewater treatment system. In particular, the bio-flocculation of microalgae and the subsequent self-settling recovery, among other benefits derived from the synergistic interaction, has attracted serious attention [4,7–9]. In previous studies, the

best experimental outcomes regarding removal potential have been reported largely for MAS inoculum ratios of 1:3 [10–12], 3:1 [13], 1:5, and 1:2 [14,15], using both high-rate algae pond (HRAP) and photobioreactors (PBRs). The MAS inoculum ratio has been shown to significantly influence pollution removal efficiency in co-culture systems [16]. However, when it comes to concentrations of inoculum ratio, there is limited information about the implication of varying concentrations of MAS with respect to a particular inoculum ratio (e.g., 0.10:0.20 g $L^{-1}$, 0.30:0.60 g $L^{-1}$, and 0.50:1.00 g $L^{-1}$ for microalgae and activated sludge, respectively, at a fixed ratio of 1:2) on treatment efficacy and biomass production. Therefore, examining the influence of different MAS inoculum concentrations of adjudged best-performing inoculum ratio on biomass growth and overall pollutant removal may aid the improvement of treatment efficiency and further understanding of the treatment process.

In most instances, the exploration of the potential of the synergistic interaction of microalgae and activated sludge bacteria co-culture has been carried out under laboratory conditions, with a constant supply of light and manipulation of other growth factors such as temperature and pH, using synthetic wastewater and real wastewater [17–20]. The few instances that explored outdoor solar radiation for the treatment of municipal wastewater with microalgae and activated sludge in PBRs focused on the influence of seasonal variations on treatment efficacy [21], inoculum ratio on the microbial community [22], and photo-oxygenation nitrogen-N removal from biosolids dewatered centrate [23]. Ref. [24] also examined the HRAP removal efficiency of micropollutants in a pilot operation under tropical conditions using only microalgae consortia for the HRAP treatment process; however, the initial microalgae inoculum proportion of the system was not defined. All of these point to the fact that the exploration of tropical weather conditions for the cultivation of microalgae for the treatment of municipal wastewater and recovery of nutrients is scarcely documented [25]. Considering that adequate provision of light intensity and temperature constitute significant parts of the conditions necessary for the stability of co-culture systems [16], natural outdoor exploration under tropical conditions could possibly be environmentally sustainable [25].

For a more realistic simulation of wastewater treatment with respect to gaining insights into the outdoor performance and treatment efficiency of MAS, it becomes imperative to examine the biotreatment potential of MAS at different inoculum concentrations utilizing real municipal wastewater under natural outdoor conditions, with solar energy as a source of light. This would mean non-dependence on artificial lighting, and thus lead to a reduction in operational capital costs. Besides the cost-saving benefits associated with treatment in a controlled environment, which would enhance the sustainability of the process since solar energy is cheap, easily available, and contains the spectral quality of light (400–700 nm) needed for microalgae growth [26], it will provide insights into the appropriate MAS concentration that can promote the achievement of optimum treatment results. It is noteworthy to mention the plausible implication of environmental weather variability for outdoor treatment efficiency [24,27]. Although this was partially demonstrated in the lab-scale study that was conducted by [21] in China under the influence of externally supplied aeration, there is a need for a more comprehensive study of the weather seasonality effect on outdoor treatment processes, particularly in a tropical environment.

Therefore, this current study presented a novel idea that evaluated the performance of varying concentrations of microalgae-activated sludge (MAS) at a constant inoculum ratio of 1:2 in laboratory-scale photobioreactors (*n* = 18). A real anaerobically digested effluent, with ideal physicochemical and microbiological properties, was used as a substrate for the experiment under natural outdoor conditions with solar energy as a source of light. This would present more realistic insights and enhance the implementation of the MAS process for treating municipal wastewater, and also promote the adoption of circular economy practices. The treatment performance was determined based on (a) biomass growth, (b) nutrient removal, (c) removal of total coliforms and *Escherichia coli*, and (d) the suitability of the treated effluent for discharge to the environment.

## 2. Materials and Methods

### 2.1. Anaerobically Digested Municipal Wastewater

Anaerobically digested municipal wastewater obtained from an up-flow anaerobic sludge blanket (UASB) reactor of a municipal Wastewater Treatment Plant (WWTP) was used as a substrate for culturing microalgae-activated sludge (MAS). The WWTP is located in Bauru City, São Paulo, Brazil, and sits on 6683.14 square meters of space (22°16′01.0″ S, 49°05′12.05″ W). It serves a population of between 30,000 and 50,000 inhabitants at an average flow rate of 78 L s$^{-1}$. The operating temperature ranged between 17 and 25 °C, at 8 h hydraulic retention time (HRT), which is in agreement with [28]. The wastewater was stored for 1 h under room conditions to allow sedimentation of particles at the laboratory of São Paulo State University (UNESP), School of Engineering, Bauru, before being transferred to the photobioreactors (PBRs). This was performed to reduce interference with light admissibility in the PBRs. The characteristics of the anaerobically digested municipal wastewater are presented in Table 1.

**Table 1.** Mean concentration and standard deviation of substrate wastewater characteristics, being that: COD = chemical oxygen demand; TDN = total dissolved nitrogen; TDP = total dissolved phosphorus; DO = dissolved oxygen; and TSS = total suspended solids.

| Parameter | Unit | Average Value |
|---|---|---|
| pH | - | $7.00 \pm 0.04$ |
| COD | mg L$^{-1}$ | $119.0 \pm 4.70$ |
| TDN | mg N L$^{-1}$ | $58.2 \pm 0.60$ |
| TDP | mg PO$_4^{3-}$ L$^{-1}$ | $6.00 \pm 0.20$ |
| DO | mg O$_2$ L$^{-1}$ | $0.59 \pm 0.25$ |
| TSS | g TSS L$^{-1}$ | $0.07 \pm 0.05$ |
| Total Alkalinity | mg CaCO$_3$ L$^{-1}$ | $339.30 \pm 19.50$ |
| Volatile Fatty Acid | mg L$^{-1}$ | $61.30 \pm 8.50$ |

*n* = 3.

### 2.2. Microorganisms and Culture Condition

Native microalgae grown in UASB anaerobically digested municipal wastewater, with 1.69 g L$^{-1}$ of total suspended solids (TSS), was used as culture inoculum. The microalgae inoculum was a mixed community of *Chlorella* sp. (65.4%), *Cyanobium* (13.6%), *Desmosdesmus* (8.1%), *Chlamydomonas* (7.4%), and *Tetradesmus* (5.4%). The microalgae were morphologically identified at the genus level based on specialized studies, using microalgae databases [29] and identification keys [30]. Samples of 2 mL were collected and fixed with formalin at a concentration of 5% [30].

The activated sludge was obtained from a WWTP located at Botucatu City, São Paulo, Brazil, and the TSS was found to be 25.76 g L$^{-1}$. Prior to use, the activated sludge was incubated in a 10 L polyethylene reactor at a temperature range of 24 to 30 °C under a light: dark photoperiod of 12:12 h at 154 ± 8 mmol m$^{-2}$ s$^{-1}$ for 14 days, with daily wastewater (WW) replacement (gradually increased the operational municipal anaerobic WW in mixture with the WW from the Botucatu WWTP) for acclimatization. Then, the respective concentrations of the varying proportions of inoculum ratio 1:2 were determined according to Table 2 and operated in batch mode under natural outdoor conditions. A Minjiang pump PS 950 with a flow rate of 0.5 L min$^{-1}$ with sparging stones was used for continuous stirring and oxygen supply. MAS and microalgae inoculum concentrations for experimental conditions and controls were cultured in 1.465 L anaerobically digested wastewater, to keep within the 2 L operational capacity mark.

**Table 2.** Composition of operational volume of microalgae and activated sludge per condition and control.

| | Conditions | Volume of Microalgae Inoculated | Volume of Activated Sludge Inoculated |
|---|---|---|---|
| 1 | Microalgae (0.10 g L$^{-1}$) + Activated sludge (0.20 g L$^{-1}$) | 0.118 L | 0.016 L |
| 2 | Microalgae (0.25 g L$^{-1}$) + Activated sludge (0.50 g L$^{-1}$) | 0.296 L | 0.039 L |
| 3 | Microalgae (0.40 g L$^{-1}$) + Activated sludge (0.80 g L$^{-1}$) | 0.473 L | 0.062 L |
| | Controls | | |
| 4 | Microalgae (0.10 g L$^{-1}$) | 0.118 L | --- |
| 5 | Microalgae (0.25 g L$^{-1}$) | 0.296 L | --- |
| 6 | Microalgae (0.40 g L$^{-1}$) | 0.473 L | --- |

Concentrations of microalgae (1.69 g L$^{-1}$) and activated sludge (25.76 g L$^{-1}$) inoculum.

## 2.3. Experimental Setup

The experimental setup consists of 18 laboratory-scale Duran® bottle photobioreactors (three conditions each for the experimental and control groups, considering the triplicate), each with a 2.2 L capacity (27 cm length and 13 cm diameter), and was operated within a 2 L capacity.

Conditions 1 to 3 (Table 2) were inoculated with different concentrations of microalgae-activated sludge (MAS) at an inoculum ratio of 1:2, and conditions 4, 5, and 6 (control) were inoculated with the corresponding microalgae concentrations of conditions 1, 2, and 3, respectively (Figure 1). The experiment was conducted for 5 days, determined by 90% removal of dissolved phosphorus, which is considered a growth-limiting nutrient level for microalgae [31]. Equal exposure of cultures to solar energy was achieved using sparging-induced agitation [32].

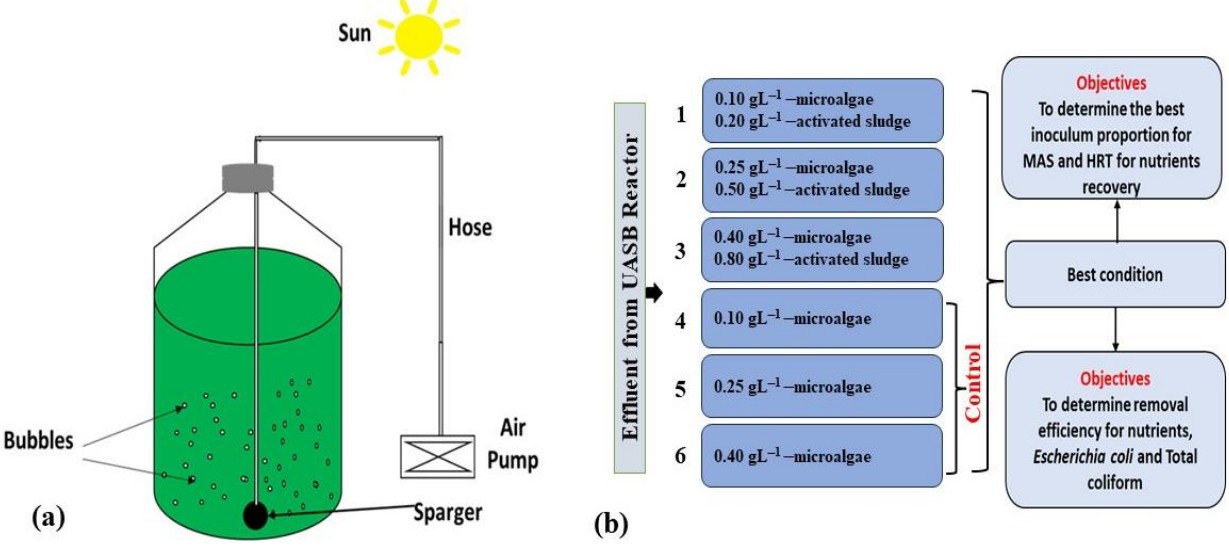

**Figure 1.** (**a**) Schematic diagram of Duran® bottle photobioreactor set-up, (**b**) composition of inoculum ratio for each condition evaluated.

### 2.4. Ambient Temperature and Light Intensity for MAS Cultivation

The experiment was conducted during the winter weather season in Brazil (27 June to 2 July 2022). The ambient temperature and light intensity readings were obtained from the automatic weather forecast located at latitude $22°21'27.6''$ and longitude $49°01'40.8''$, through the Meteorology Center at UNESP in Bauru (IPMet). The obtained data were recorded at 5 min intervals, and the solar radiation data were converted from $R_s$ ($W\,m^{-2}$) to the photosynthetically active radiation (PAR) ($\mu mol\,m^{-2}\,s^{-1}$) using the conversion factor of 2.02, according to [33]. Therefore, the daily average light intensity ranged from 694.29 to 841.31 $\mu mol\,m^{-2}\,s^{-1}$, and the temperature ranged from 18.0 to 20.0 °C, implying that the MAS culture was grown in a moderate temperature range. Hence, the light: dark photoperiod averaged 11.5:12.5 h per day during the period of the experiment.

### 2.5. Analytical Methods

The total biomass growth pattern in cultures was evaluated daily with TSS (2540 D, [34]), while the optical density was measured at a wavelength of 680 nm ($OD_{680nm}$) using a NANOCOLOR UV/VIS II Spectrophotometer, and microalgae cell count (cell $mL^{-1}$) was determined with an optical microscope from Leica microsystems, D. 35578, Wetzlar, Germany. Daily monitoring of nutrient removal (total dissolved phosphorus (TDP) and total dissolved nitrogen (TDN)) was performed using a Spectrophotometer NANO-COLOR UV/VIS II (4500-P-E for TDP, 4500-$NO_3$-B, 4500-$NO_2$-B, 4500-$NH_3$-B, and C for TDN; [34]) ($n = 18$). Total alkalinity, determined by titration potentiometric with sulfuric acid (2320-B; [34]), was evaluated on day 0 ($D_0$), day 3 ($D_3$), and day 5 ($D_5$) of the experiment. Total coliforms and *E. coli* were determined on $D_0$ and $D_5$ of the experiment using Chromocult Coliform Agar (Pour Plate 9215B, Merck KGaA, Darmstadt, Germany, [34]) in triplicate. The pH (4500-HB, [34]) was monitored daily ($n = 18$).

The nutrients analyzed were the dissolved nutrients (nitrate, nitrite, ammonium, and phosphate), which were determined according to [34]. The samples were first filtered through 1.2 $\mu m$ pore size glass fiber membrane filters and then through 0.45 $\mu m$ pore size cellulose acetate membrane filters to determine the dissolved nutrients. Additionally, total suspended solids (TSS) gravimetric analysis for productivity (dry weight) represented the biomass in the culture medium.

### 2.6. Statistical Analysis

The results were entered into an Excel spreadsheet, and SPSS Version 22.0 was employed for the statistical analysis. Data were analyzed for mean, analysis of variance (ANOVA), and least significant difference (LSD) post hoc tests. A t-test was used to determine variations in the concentrations of parameters analyzed between $D_0$ and $D_5$, with a 5% level of significance. The relationship between microalgal-activated sludge (MAS) bacteria inoculum and nutrient recovery was investigated using Pearson's correlation analysis. The geometric mean was used to calculate the average total coliform and *E. coli* bacteria population to avoid probable distortion from the varying triplicate values [35].

## 3. Results and Discussion

### 3.1. Outdoor Temperature and Light Intensity: Potential for Biomass Growth

The experiment was conducted at a moderate temperature range (18.0 to 20.0 °C), based on the optimum temperatures reported in the literature for microalgal growth, but within the interval considered suitable for photosynthetic efficiency and growth for most microalgae species [36–38]. For light intensity (694.29 to 841.31 $\mu mol\,m^{-2}\,s^{-1}$), the range of values recorded is considered adequate [38,39].

The potential for increased growth with higher light intensity was demonstrated after a gradual increase from 600 to 1500 $\mu mol\,m^{-2}\,s^{-1}$ yielded a significant percentage growth difference (~92%, dry weight) in [39]. The positive growth response was possibly due to the penetrative strength of the new light intensity, which restrained the shading of the biomass that may have limited the photosynthetic response of microalgae at 600 $\mu mol\,m^{-2}\,s^{-1}$.

This suggests the possibility of increased biomass productivity in the summer weather season in an outdoor treatment situation.

This evaluation becomes essential as temperature and light intensity values play crucial roles in the growth and productivity of microalgae culture for an outdoor photo-bioreactor treatment, in addition to nutrient adequacy and appropriate mixing [40,41].

### 3.2. Operational and Environmental Conditions

The differences in pH values between the experimental conditions are not statistically significant ($p > 0.05$; Table 3). An increase in pH from near neutral to alkaline was observed on the second day of the experiment, and the pH remained in the alkaline zone until the end of the batch experiment. Ref. [42] reported similar findings in reactors without pH control during the evaluation of different pH levels on microalgae cultivation and biomass recovery that was operated in batch mode. The pH values differed in the order $1 > 2 > 3$ (Table 3), which shows different levels of $CO_2$ fixations through photosynthesis in the PBRs, although not significantly ($p > 0.5$) different. pH values for controls also differed in the order $4 < 5 < 6$, with no significant difference.

**Table 3.** Mean and standard deviation of pH, total alkalinity, biomass productivity, and cell density at optical density 680 nm, and nutrient removal efficiencies found in the different conditions.

| Condition | pH | Total Alkalinity (mg CaCO$_3$ L$^{-1}$) | ˆ Total Biomass Productivity (g TSS L$^{-1}$ d$^{-1}$) | ˆ Cell Density (OD$_{680nm}$) | TDP Removal (%) | TDN Removal (%) |
|---|---|---|---|---|---|---|
| 1 | 9.40 ± 1.30 [a] | 163.70 ± 73.70 | 0.10 ± 0.01 [a] | 0.84 ± 0.10 [a] | 85.1 ± 1.04 | 66.1 ± 6.40 |
| 2 | 9.40 ± 1.20 [a] | 211.40 ± 63.30 | 0.05 ± 0.02 [b] | 0.31 ± 0.03 [b] | 40.7 ± 10.30 | 16.4 ± 5.80 |
| 3 | 8.90 ± 0.90 [a] | 235.30 ± 87.00 | 0.04 ± 0.03 [b] | 0.17 ± 0.10 [b] | −43.7 ± 15.70 | −62.90 ± 10.04 |
| Control | | | | | | |
| 4 | 9.40 ± 1.30 [a] | 194.80 ± 49.00 | 0.09 ± 0.01 [a] | 0.75 ± 0.30 [a] | 83.9 ± 10.40 | 43.20 ± 13.60 |
| 5 | 9.50 ± 1.20 [a] | 211.80 ± 53.80 | 0.11 ± 0.03 [a] | 0.97 ± 0.01 [a] | 85.5 ± 8.80 | 58.3 ± 7.00 |
| 6 | 9.60 ± 1.20 [a] | 205.30 ± 68.70 | 0.13 ± 0.02 [a] | 1.10 ± 0.02 [a] | 92.3 ± 1.20 | 60.6 ± 5.10 |

$n = 18$; ˆ Mean difference (D$_0$–D$_5$); [a,b] Means without a common superscript letter in a column differ ($p < 0.05$) as analyzed by one-way ANOVA and LSD. TDP and TDN indicate the dissolved proportion of Phosphorus-*p* and Nitrogen-N, respectively, in the solution as samples were filtered through 0.45 μm membrane pore size prior to determination.

The co-culture of microalgae and activated sludge has been shown to affect the level of pH in the reactor with respect to the conversion of nitrogenous compounds [43]. Removal of nitrogen in the UASB reactor is negligible but becomes mineralized ($NH_4^+$-N) through the hydrolysis of protein and urea [24,44]. Resultantly, $NH_4^+$-N was the predominant form of nitrogen in the substrate wastewater. Considering ammonium ($NH_4^+$-N) is easily assimilated as the preferred form of nitrogen by microalgae with less energy dissipation [45], nitrogen removal was considerably largely by assimilation and followed by nitrification, and as such may have contributed less to increasing pH values from slightly acidic to alkaline. According to [16], the co-culture systems synchronously achieve the removal of nitrogen and chemical oxygen demand (COD) by microalgae and activated sludge, respectively.

In previous studies, the pH increase was attributed to a reduction in bicarbonates in the form of $CO_2$ fixation in the cultures, enabled by photosynthesis [46,47]. This invariably leads to a reduction in total alkalinity concentration [46–49], which serves as a source of inorganic carbon for growth. A similar occurrence was also observed in this experiment. In Table 3, deviation reflects alkalinity reduction across the conditions (not statistically significant, $p = 1.43$), which is an indication of the consumption of inorganic carbon in the form of carbonate and bicarbonate present in the effluent [50].

### 3.3. Effect of Inoculum Concentrations on Total Biomass Productivity

Total biomass productivity (microalgae cells and sludge bacteria) was evaluated with respect to different MAS inoculum concentrations on photosynthetic responses and

subsequent microalgae cell growth and bacterial cell replication. Cell productivity varied significantly ($p = 0.001$) among the experimental cultures. The LSD post hoc tests revealed condition 1 was significantly higher than conditions 2 and 3, and fairly within the same range for the controls (conditions 4–6; $p > 0.05$; see Tables 3, A1 and A2). The proportions of activated sludge inoculated in conditions 2 and 3 appeared to be in excess of the capacity of the systems and thus compromised the performance by interfering with photosynthesis and the subsequent productivity rate in the reactors.

Therefore, the MAS inoculum concentration for wastewater treatment with respect to the treatment capacity of the system is critical to the translucence of cultures in PBRs and consequently the light energy received from solar radiation, which is required for replication and growth of microalgae, thus probably limiting oxygen supply within the systems to sludge bacteria due to altered gaseous exchange involving MAS [51–53]. This shows that while light intensity is essential for microalgae productivity [54,55], the MAS concentration of cultures seems to play an essential role in the amount of irradiance (light received), the energy available for photosynthesis, and the subsequent gaseous exchange for optimum synergistic performance of the systems to achieve effective treatment.

Notably, the results of productivity for condition 1 and the control experiment (4, 5, and 6) were similar to the values in the existing literature for microalgae cultivations conducted outdoors [56,57]. Ref. [56] reported 0.140 g TSS $L^{-1}$ $d^{-1}$ productivity in a large-scale cultivation of *Coccomyxa onubensis* with a synthetic commercial NPK fertilizer solution that was operated for 30 days outdoors. [57] also reported a productivity range of 0.09 to 0.19 g TSS $L^{-1}$ $d^{-1}$ in the pilot study that cultivated *Tetradesmus obliquus* and *Graesiella emersonii* in local tap water for 9 months. In our study, native microalgae and activated sludge (MAS) in different concentrations (three conditions) were cultivated in real anaerobically digested municipal wastewater in laboratory-scale PBRs outdoors. We evaluated the best MAS concentration and obtained biomass productivity 0.100 g TSS $L^{-1}$ $d^{-1}$, 0.090 g TSS $L^{-1}$ $d^{-1}$, 0.110 g TSS $L^{-1}$ $d^{-1}$, and 0.130 g TSS $L^{-1}$ $d^{-1}$ for conditions 1, 4, 5, and 6, respectively.

Considering that the total of the TSS did not always refer to biomass, since there may be a significant percentage of inorganic solids, the estimation of microalgae cell growth and density was also determined in terms of $OD_{680nm}$. The results of measurements followed the same trend that was noted for productivity (Table 3). The strong positive correction that was established between the models predicted cell growth and estimated cultured growth at $OD_{680nm}$ for a mixed culture (microalgae and bacteria) and evidently demonstrated cell growth measurement potential for MAS at $OD_{680nm}$ [58].

As previously noted, conditions 2 and 3 (high MAS inoculum concentrations) (Table 2) had limited cell growth, as evidenced by low cell growth estimates at $OD_{680nm}$. A potential explanation for this is the die-off of microalgae and subsequent cell ruptures due to photoinhibition resulting from shading [53]. Likewise, the secretion of chemical substances (algicidal) by bacteria, which decimate microalgae, and the shading effect on microalgae from bacteria could result in mass death and interfere with productivity [59]. Hence, it is important to determine the appropriate MAS inoculum concentrations for a treatment system to achieve the optimum treatment outcome.

The fragments of intracellular pigments released into the cultures from ruptured cells were considered minute and undetected at $OD_{440nm}$ and $OD_{680nm}$, as established in the sonication monitoring study of three microalgae species *(M. aeruginosa*, *C. pyrenoidosa*, and *C. reinhardtii)* using optical density estimate of microalgal suspension, intracellular pigments and proteins, and cell counting [60]. Moreover, ref. [50] have shown that biomass measurement at $OD_{682nm}$ is commonly indicative of biomass growth and not cell rupture.

Additionally, the photoautotrophic growth of microalgae cells in the cultures is shown in Figure 2. No lag phase was observed except for condition 3 (culture with highest MAS inoculum concentration: 0.40 g $L^{-1}$ of microalgae + 0.80 g $L^{-1}$ of activated sludge), with a slight decline on the second day of the experiment, suggesting die-off of some microalgae cells. Microalgae cell density varied significantly across the conditions ($p = 0.000$) and

ranged between $1.99 \times 10^6$ and $1.60 \times 10^7$ cells mL$^{-1}$, $5.30 \times 10^6$ and $1.30 \times 10^7$ cells mL$^{-1}$, and $1.07 \times 10^7$ and $1.83 \times 10^7$ cells mL$^{-1}$ for conditions 1 to 3, respectively. Among the cultures, conditions 2 and 3 decreased in cell density on day 4 of the experiment as evidenced by the growth trend (Figure 2), suggesting growth of microalgae cells is limited by irradiance from the shading effect [59], which could mean that the inoculum concentrations were beyond what the systems are capable of accepting to achieve effective treatment. This could be attributed to poor translucence caused by bacteria shading and linked to the proportions of MAS inoculum concentrations which are probably in excess of what the systems can accommodate to initiate optimum cell growth and adequate treatment [53,59].

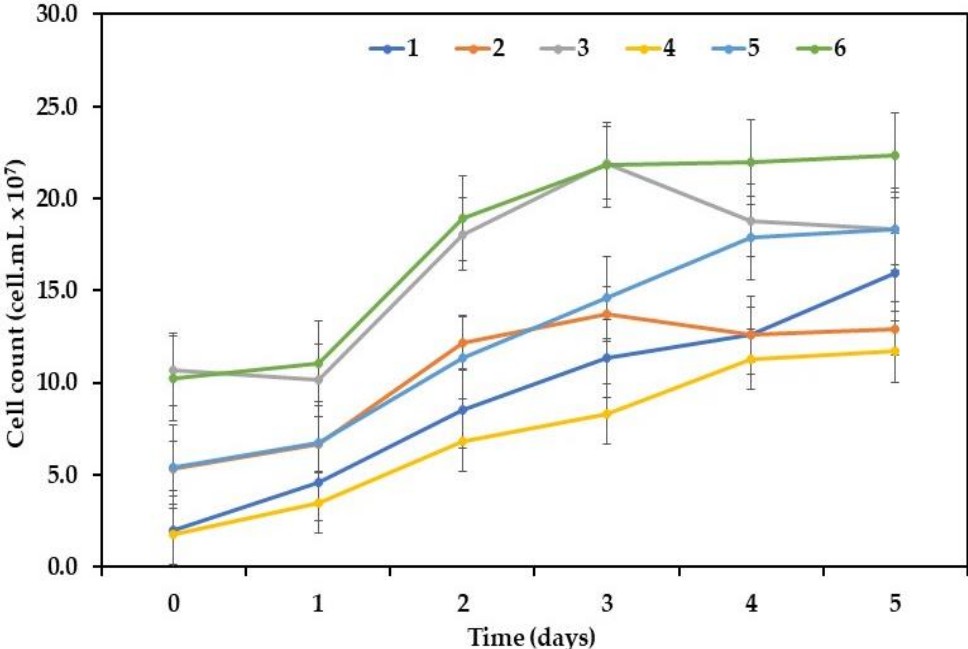

**Figure 2.** Average cell count expressed in cell mL$^{-1}$ for the conditions.

As indicated above, in condition 1 among the experimental scenarios, and similarly for controls 4, 5, and 6, the microalgae cell growths were similar to the findings of [61]. They cultivated native microalgae (predominantly *Scenedesmus*) in municipal wastewater using a pilot-scale photobioreactor outdoors, and achieved maximum mixotrophic ($1.887 \times 10^7$ cells mL$^{-1}$) and heterotrophic ($1.473 \times 10^7$ cells mL$^{-1}$) cell production. Similarly, in the study conducted by [62], where *T. weissflogii* was cultivated outdoors all year round in batches using seawater-enriched medium (f/2 medium) substrate in a transparent poly-methyl methacrylate photobioreactor, the obtained cell growth results (June–August) aligned with the results obtained for conditions 1, 4, 5, and 6 of this present study. It is expedient to mention that inoculum concentrations affected the growth of microalgae cells, with the lowest inoculum proportions yielding higher cell growth (1 > 2 > 3), except for condition 5 among the controls that yielded higher microalgae cell growth than (6 > 4). After attaining a certain level during the experiment, particularly the controls, cell growth tends to decrease and actually decreased in the case of conditions 2 and 3. Optical limitation or photoinhibition induced by too high or too low inoculum concentrations could affect cell growth [63]. The amount of initial inoculum also has a significant influence on cell productivity and growth rate, with the lowest initial inoculum producing higher biomass density [63,64]. Nonetheless, a definitive comparison is challenging because of limited information and the varying cultivation techniques employed.

*3.4. Effect of Inoculum Concentrations on Nutrient Removal*

The end of the experiment was premised on the attainment of $\geq$90% TDP removal (mg P-PO$_4$ L$^{-1}$), beyond which further significant growth levels for microalgae cells may not be achieved [31]. Therefore, 92.3% of TDP uptake was attained in condition 6 (control group) at D$_5$ and 85.1% of TDP uptake was attained in condition 1 among the experiment group, with no significant difference ($p > 0.05$).

The percentage removal of total dissolved phosphorus (TDP) and total dissolved nitrogen (TDN) varied significantly among the three conditions ($p < 0.05$). The TDP (85.1%) and TDN (66.1%) removal were highest in condition 1. Considering the control conditions (4 to 6), condition 6 had the highest TDP removal (92.3%) (Table 3). Based on the results obtained in nutrient removal, there were observable increases in the uptake of TDP and TDN in conditions 1 and 2, and the controls (4, 5, and 6), under different MAS and microalgae inoculum concentrations, while the concentrations of TDP and TDN increased by 43.7% and 62.9%, respectively, in condition 3. This is a reflection of the decrease in cell count observed on the second day of the experiment for condition 3 (Figure 2). The fraction of activated sludge that makes up the MAS concentration for this condition could have interfered with light penetration, and consequent microalgae cell rupture, thus releasing nutrients within the solution [13]. For the experimental conditions, the uptake of TDP and TDN in condition 1 on D$_5$, the last day of the experiment, was significantly higher than in conditions 2 and 3 ($p < 0.05$), while it was statistically similar to the controls (4, 5, and 6) (Appendix A Tables A3 and A4) which contained the proportionate concentrations of microalgae in conditions 1, 2, and 3, respectively. This suggests that the MAS inoculum concentration, particularly for activated sludge, can significantly affect the pollution abatement potential in relation to the biomass growth of co-culture systems. Information on nutrient loads of MAS and microalgae and the probable add-on to nutrient levels in the reactors from the different inoculum proportions was not provided as this was not evaluated in the study. It has been shown that optimum biomass productivity produces the best nutrient removal performance during cultivation [65–68]. This explains the strong negative relationship between optical density (OD$_{680nm}$) and TDP uptake, with obviously increased biomass production leading to reduced concentration of TDP (r = $-0.7$ for condition 1, r = $-0.8$ each for control conditions 4, 5, and 6), and conversely for conditions 2 and 3 (r = 0.5, 0.7, respectively) (Table 4).

**Table 4.** Correlation between TDP removal and OD$_{680}$ of the conditions.

| Condition/Control | TDP1 | TDP2 | TDP3 | TDP 4 | TDP5 | TDP6 |
|---|---|---|---|---|---|---|
| OD1 | −0.7 | 0.0 | 0.7 | −0.7 | −0.7 | −0.8 |
| OD2 | −0.2 | 0.5 | 0.8 | −0.4 | −0.4 | −0.4 |
| OD3 | 0.0 | 0.6 | 0.7 | −0.2 | −0.1 | −0.2 |
| Control | | | | | | |
| OD4 | −0.8 | −0.1 | 0.7 | −0.8 | −0.8 | −0.8 |
| OD5 | −0.8 | −0.2 | 0.6 | −0.8 | −0.8 | −0.8 |
| OD6 | −0.8 | −0.2 | 0.6 | −0.8 | −0.8 | −0.8 |

TDP = total dissolved phosphorus; OD = optical density; Negative: ▬ indicates strong negative relationship between microalgae growth (OD$_{680}$) and uptake of total dissolved phosphorus (TDP); Neutral: ▭ indicates there is no relationship between microalgae growth (OD$_{680}$) and uptake of total dissolved phosphorus (TDP); Positive: ▬ indicates strong possitive relationship and between microalgae growth (OD$_{680}$) and uptake of total dissolved phosphorus (TDP).

These results suggest that the concentration of MAS inoculum to be used for microalgae cultivation and bioremediation of wastewater constitutes an essential condition that should be factored into experimental set-ups, which is in agreement with previously reported findings by [19]. Therefore, beyond defining the inoculum ratio for a mix-culture of MAS, it is essential to determine the appropriate MAS concentration with respect to the operational size of PBRs, to avoid any form of interference in the photosynthetic process that can consequently limit microalgae cell growth and replication.

The obtained results clearly showed the feasibility of nutrient assimilation by microalgae-based systems in an outdoor environment when the appropriate inoculum concentration is used. This is without prejudice to other growth-influencing factors such as temperature, light intensity, and pH. Overall, this presents potential benefits for nutrient cycling in an efficient manner, making them usable as bio-fertilizers for the production of food and cash crops [69–72].

### 3.5. Assessment of Total Coliforms and Escherichia coli Removal

The potential of MAS to remove total coliforms and *E. coli* from municipal wastewater was evaluated on $D_0$ and $D_5$ (Table 5) and Logarithmic removal (Log-Re) performance is shown in Figure 3. Complete Log-Re was achieved for *E. coli* in all the experiments tested (conditions 1 to 3 and controls 4 to 6).

**Table 5.** Geometric mean of total coliforms and *E. coli* at $D_0$ and $D_5$ in the conditions.

| Conditions/ Controls | Day 0 ($D_0$) | | Day 5 ($D_5$) | |
|---|---|---|---|---|
| | Total Coliform (CFU 100 mL$^{-1}$) | *Escherichia coli* (CFU 100 mL$^{-1}$) | Total Coliform (CFU 100 mL$^{-1}$) | *Escherichia coli* (CFU 100 mL$^{-1}$) |
| 1 | $2.79 \times 10^5$ | $3.59 \times 10^4$ | $1.07 \times 10^2$ | $0.00 \times 10^0$ |
| 2 | $1.48 \times 10^5$ | $3.55 \times 10^4$ | $5.23 \times 10^2$ | $0.00 \times 10^0$ |
| 3 | $1.83 \times 10^5$ | $4.33 \times 10^4$ | $8.81 \times 10^2$ | $0.00 \times 10^0$ |
| Controls | | | | |
| 4 | $1.64 \times 10^5$ | $3.79 \times 10^4$ | $0.00 \times 10^0$ | $0.00 \times 10^0$ |
| 5 | $1.77 \times 10^5$ | $3.46 \times 10^4$ | $0.00 \times 10^0$ | $0.00 \times 10^0$ |
| 6 | $1.15 \times 10^5$ | $2.91 \times 10^4$ | $0.00 \times 10^0$ | $0.00 \times 10^0$ |

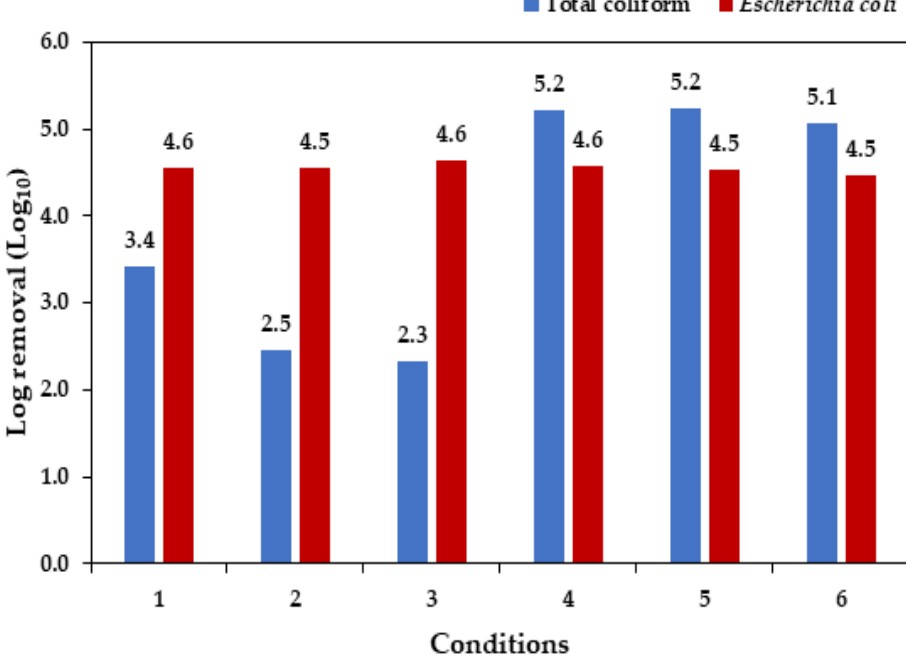

**Figure 3.** Log removal (Log-Re) for total coliforms and *E. coli*.

The control experiments had better total coliforms results, with Log-Re 5.2 in conditions 4 and 5, and 5.1 in condition 6. Moreover, total coliforms in the condition 1 system were reduced by 3.4 log units, while conditions 2 and 3 had 2.5 and 2.3 Log-Re for total coliforms, respectively.

While the complete Log-Re for *E. coli* cut across the three conditions, the lowest Log-Re for total coliforms was observed in conditions 2 and 3, and varied significantly from condition 1. This is probably a result of the low microalgae biomass growth in these

systems, with consequences for nutrient availability and reduced secretion of antibacterial compounds [73–75]. Additionally, the operational conditions of cultures, such as increased pH and competition with bacteria, may have contributed to the inactivation of pathogenic microorganism indicators (*E. coli*).

For *E. coli*, a high die-off rate was reported in the alkaline medium [76,77]. This was probably responsible for the outright Log-Re in the *E. coli* population in all the cultures on $D_5$, the last day of the experiment. The results obtained were comparable with the findings from [76,77]. Importantly, the result showed the potential of the resultant wastewater to be deployed for non-potable purposes such as irrigation and lawn watering without posing a risk of contamination by pathogenic organisms, and complies with the World Health Organization standard of $10^3$ MPN 100 mL$^{-1}$ for *E. coli* for non-potable purposes [78].

### 3.6. MAS Inoculum Concentration for Wastewater Treatment

The mean difference between $D_0$ and $D_5$ was calculated for selected parameters using a 95% confidence interval. From Table 6, the obtained significant variation ($p < 0.05$) further confirmed the previously mentioned inoculum concentration that fostered biomass productivity, nutrient removal efficiency, and high recovery potential.

**Table 6.** *p*-value of mean difference for D0 and D5 of selected parameters at 95% confidence interval.

| Parameter | Condition | | | | | |
|---|---|---|---|---|---|---|
| | 1 | 2 | 3 | 4 | 5 | 6 |
| Total Alkalinity | 0.0177 | 0.0322 | 0.0191 | 0.0320 | 0.0119 | 0.0054 |
| Productivity | 0.0033 | 0.1475 * | 0.1711 | 0.0015 | 0.0260 | 0.0067 |
| OD$_{680}$ | 0.0041 | 0.0041 | 0.5139 | 0.0384 | 0.0001 | 0.0001 |
| TDP | 0.0010 | 0.0217 | 0.2012 | 0.0166 | 0.0077 | 0.0002 |
| T. coliform | 0.006 | 0.1240 | 0.0522 | 0.0041 | 0.0002 | 0.0003 |

* Values in red font color are not statistically significant. Productivity = total biomass productivity OD$_{680}$ = optical density at 680 nm, TDP = total dissolved phosphorus, T. coliform = total coliforms.

Among the experimental conditions, condition 1 (0.10/0.20 g L$^{-1}$) demonstrated the potential for use in an outdoor treatment system for wastewater and the recovery of high-value products, drawing from the high performance of the MAS inoculum in the treatment systems, as against others which seem to undermine the treatment capacity of the systems. This removal depicts a promising alternative means of providing economically viable resources in an environmentally friendly manner while supporting the resilience of overburdened natural systems [79,80].

The overriding importance of this optimum inoculum concentration, besides enabling treatment efficacy and nutrient uptake, stems from the potential for high biomass recovery through self-settling. The synergistic interaction of MAS aids the formation of bio-flocculation and the recovery of high-quality and nutrient-rich biomass by sedimentation under gravity [4]. The bio-flocs are made of extracellular polymeric substances (EPS) separately produced by microalgae and bacteria, which are subsequently bonded together and thus mediate the formation of aggregates of efficient settling characteristics by gravity [81], yielding excellent biomass harvest outcomes.

### 4. Conclusions

Evaluating the performance of microalgae and activated sludge (MAS) using different inoculum concentrations for treating anaerobically digested municipal wastewater, it can be stated that inoculum concentration influenced the performance of MAS in terms of total biomass growth and nutrient uptake.

It was shown that a MAS inoculum concentration of 0.10/0.20 g L$^{-1}$ (total biomass productivity: 0.10 g TSS L$^{-1}$ d$^{-1}$, TDP: 85.1%, and TDN: 66.1%) significantly out-performed 0.20/0.40 and 0.40/0.80 g L$^{-1}$, possibly due to photosynthetic interference by the proportions of activated sludge in the latter and optical limitation from excess initial inoculum concentration beyond the capability of the treatment reactors.

For the removal of total coliforms and *E. coli*, while outright Log-Re was recorded for *E. coli*, influenced by the high pH value in the PBRs, across the conditions, the lowest Log-Re was recorded in 0.20/0.40 and 0.40/0.80 g $L^{-1}$ MAS inoculum concentration for total coliforms, suggesting that increased inoculum concentration of MAS may have a negative effect on treatment efficacy.

In general, to derive optimum benefit from the synergy of MAS inoculum for wastewater treatment and recovery of biomolecules, the deployment of appropriate inoculum proportion is germane. However, conducting this experiment on a pilot scale will alleviate concerns about the feasibility of deployment in a real-life scenario.

**Author Contributions:** Conceptualization and methodology, A.O.J., G.R. and G.H.R.S.; Laboratory analysis, data curation, and statistical analysis, A.O.J., A.O.B., C.M.E.P., and G.A.S.A.D.; manuscript draft preparation, A.O.J.; manuscript review and editing, A.O.B., C.M.E.P., G.H.R.S. and G.R.; supervision, G.H.R.S. and G.R.; funding acquisition and project administration, G.H.R.S. All authors have read and agreed to the published version of the manuscript.

**Funding:** This paper was possible thanks to the scholarship granted from the Brazilian Federal Agency for Support and Evaluation of Graduate Education (CAPES), in the scope of the Program CAPES-PrInt, process number 88887.310463/2018-00, and CAPES–Finance code 001; National Council for Scientific and Technological Development (CNPq) processes 309064/2018-0, 308663/2021-7, and 427936/2018-7, and the Sao Paulo Research Foundation (FAPESP) processes 2018/18367-1 and 2022/07475-3. In addition, the Tertiary Education Trust Fund (TETFund) and the Forum for Agricultural Research in Africa (FARA) (award number: TETF/DASTD/TSAS/MOU/FARA/2020/VOL.I) provided support for undertaking a master's degree program at Sao Paulo State University (UNESP), Brazil.

**Institutional Review Board Statement:** Not applicable.

**Informed Consent Statement:** Not applicable.

**Data Availability Statement:** The data presented in this study are available on request from the corresponding author.

**Acknowledgments:** We appreciate the support received from Glauco Perpetuo, Thalita Lacerda, and Eduardo Miguel during the laboratory analyses of this study. We also appreciate the UNESP IPMet for the gracious and prompt supply of weather data.

**Conflicts of Interest:** The authors declare no conflict of interest.

## Appendix A Least Significant Post Hoc Test for Total Biomass Productivity Estimates, and Percentage of Total Dissolved Phosphorus and Nitrogen Uptake

**Table A1.** Dependent Variable: Total Biomass Productivity.

| LSD | | Mean Difference (I–J) | Std. Error | Sig. | 95% Confidence Interval | |
| --- | --- | --- | --- | --- | --- | --- |
| Conditions | | | | | Lower Bound | Upper Bound |
| con_1 | con_2 | −0.09517 * | 0.02806 | 0.001 | −0.1508 | −0.0395 |
| | con_3 | −0.13906 * | 0.02806 | 0.000 | −0.1947 | −0.0834 |
| | con_4 | 0.05206 | 0.02806 | 0.066 | −0.0036 | 0.1077 |
| | con_5 | 0.01828 | 0.02806 | 0.516 | −0.0374 | 0.0739 |
| | con_6 | −0.02783 | 0.02806 | 0.324 | −0.0835 | 0.0278 |
| con_2 | con_1 | 0.09517 * | 0.02806 | 0.001 | 0.0395 | 0.1508 |
| | con_3 | −0.04389 | 0.02806 | 0.121 | −0.0995 | 0.0118 |
| | con_4 | 0.14722 * | 0.02806 | 0.000 | 0.0916 | 0.2029 |
| | con_5 | 0.11344 * | 0.02806 | 0.000 | 0.0578 | 0.1691 |
| | con_6 | 0.06733 * | 0.02806 | 0.018 | 0.0117 | 0.1230 |

**Table A1.** *Cont.*

| LSD | | Mean Difference (I–J) | Std. Error | Sig. | 95% Confidence Interval | |
|---|---|---|---|---|---|---|
| Conditions | | | | | Lower Bound | Upper Bound |
| con_3 | con_1 | 0.13906 * | 0.02806 | 0.000 | 0.0834 | 0.1947 |
| | con_2 | 0.04389 | 0.02806 | 0.121 | −0.0118 | 0.0995 |
| | con_4 | 0.19111 * | 0.02806 | 0.000 | 0.1355 | 0.2468 |
| | con_5 | 0.15733 * | 0.02806 | 0.000 | 0.1017 | 0.2130 |
| | con_6 | 0.11122 * | 0.02806 | 0.000 | 0.0556 | 0.1669 |
| con_4 | con_1 | −0.05206 | 0.02806 | 0.066 | −0.1077 | 0.0036 |
| | con_2 | −0.14722 * | 0.02806 | 0.000 | −0.2029 | −0.0916 |
| | con_3 | −0.19111 * | 0.02806 | 0.000 | −0.2468 | −0.1355 |
| | con_5 | −0.03378 | 0.02806 | 0.231 | −0.0894 | 0.0219 |
| | con_6 | −0.07989 * | 0.02806 | 0.005 | −0.1355 | −0.0242 |
| con_5 | con_1 | −0.01828 | 0.02806 | 0.516 | −0.0739 | 0.0374 |
| | con_2 | −0.11344 * | 0.02806 | 0.000 | −0.1691 | −0.0578 |
| | con_3 | −0.15733 * | 0.02806 | 0.000 | −0.2130 | −0.1017 |
| | con_4 | 0.03378 | 0.02806 | 0.231 | −0.0219 | 0.0894 |
| | con_6 | −0.04611 | 0.02806 | 0.103 | −0.1018 | 0.0095 |
| con_6 | con_1 | 0.02783 | 0.02806 | 0.324 | −0.0278 | 0.0835 |
| | con_2 | −0.06733 * | 0.02806 | 0.018 | −0.1230 | −0.0117 |
| | con_3 | −0.11122 * | 0.02806 | 0.000 | −0.1669 | −0.0556 |
| | con_4 | 0.07989 * | 0.02806 | 0.005 | 0.0242 | 0.1355 |
| | con_5 | 0.04611 | 0.02806 | 0.103 | −0.0095 | 0.1018 |

* The mean difference is significant at the 0.05 level.

**Table A2.** Dependent Variable: OD680.

| LSD | | Mean Difference (I–J) | Std. Error | Sig. | 95% Confidence Interval | |
|---|---|---|---|---|---|---|
| Conditions | | | | | Lower Bound | Upper Bound |
| con_1 | con_2 | 0.53267 * | 0.10011 | 0.000 | 0.3145 | 0.7508 |
| | con_3 | 0.67200 * | 0.10011 | 0.000 | 0.4539 | 0.8901 |
| | con_4 | 0.09300 | 0.10011 | 0.371 | −0.1251 | 0.3111 |
| | con_5 | −0.12733 | 0.10011 | 0.227 | −0.3455 | 0.0908 |
| | con_6 | −0.21867 * | 0.10011 | 0.050 | −0.4368 | −0.0005 |
| con_2 | con_1 | −0.53267 * | 0.10011 | 0.000 | −0.7508 | −0.3145 |
| | con_3 | 0.13933 | 0.10011 | 0.189 | −0.0788 | 0.3575 |
| | con_4 | −0.43967 * | 0.10011 | 0.001 | −0.6578 | −0.2215 |
| | con_5 | −0.66000 * | 0.10011 | 0.000 | −0.8781 | −0.4419 |
| | con_6 | −0.75133 * | 0.10011 | 0.000 | −0.9695 | −0.5332 |

**Table A2.** *Cont.*

| LSD | | Mean Difference (I–J) | Std. Error | Sig. | 95% Confidence Interval | |
|---|---|---|---|---|---|---|
| Conditions | | | | | Lower Bound | Upper Bound |
| con_3 | con_1 | −0.67200 * | 0.10011 | 0.000 | −0.8901 | −0.4539 |
| | con_2 | −0.13933 | 0.10011 | 0.189 | −0.3575 | 0.0788 |
| | con_4 | −0.57900 * | 0.10011 | 0.000 | −0.7971 | −0.3609 |
| | con_5 | −0.79933 * | 0.10011 | 0.000 | −1.0175 | −0.5812 |
| | con_6 | −0.89067 * | 0.10011 | 0.000 | −1.1088 | −0.6725 |
| con_4 | con_1 | −0.09300 | 0.10011 | 0.371 | −0.3111 | 0.1251 |
| | con_2 | 0.43967 * | 0.10011 | 0.001 | 0.2215 | 0.6578 |
| | con_3 | 0.57900 * | 0.10011 | 0.000 | 0.3609 | 0.7971 |
| | con_5 | −0.22033 * | 0.10011 | 0.048 | −0.4385 | −0.0022 |
| | con_6 | −0.31167 * | 0.10011 | 0.009 | −0.5298 | −0.0935 |
| con_5 | con_1 | 0.12733 | 0.10011 | 0.227 | −0.0908 | 0.3455 |
| | con_2 | 0.66000 * | 0.10011 | 0.000 | 0.4419 | 0.8781 |
| | con_3 | 0.79933 * | 0.10011 | 0.000 | 0.5812 | 1.0175 |
| | con_4 | 0.22033 * | 0.10011 | 0.048 | 0.0022 | 0.4385 |
| | con_6 | −0.09133 | 0.10011 | 0.380 | −0.3095 | 0.1268 |
| con_6 | con_1 | 0.21867 * | 0.10011 | 0.050 | 0.0005 | 0.4368 |
| | con_2 | 0.75133 * | 0.10011 | 0.000 | 0.5332 | 0.9695 |
| | con_3 | 0.89067 * | 0.10011 | 0.000 | 0.6725 | 1.1088 |
| | con_4 | 0.31167 * | 0.10011 | 0.009 | 0.0935 | 0.5298 |
| | con_5 | 0.09133 | 0.10011 | 0.380 | −0.1268 | 0.3095 |

* The mean difference is significant at the 0.05 level.

**Table A3.** Dependent Variable: Total dissolved Phosphorus uptake (%).

| LSD | | Mean Difference (I–J) | Std. Error | Sig. | 95% Confidence Interval | |
|---|---|---|---|---|---|---|
| Conditions | | | | | Lower Bound | Upper Bound |
| con_1 | con_2 | 44.42500 * | 7.76082 | 0.000 | 27.5156 | 61.3344 |
| | con_3 | 128.58200 * | 7.76082 | 0.000 | 111.6726 | 145.4914 |
| | con_4 | 1.13100 | 7.76082 | 0.887 | −15.7784 | 18.0404 |
| | con_5 | −0.41200 | 7.76082 | 0.959 | −17.3214 | 16.4974 |
| | con_6 | −7.20667 | 7.76082 | 0.371 | −24.1160 | 9.7027 |
| con_2 | con_1 | −44.42500* | 7.76082 | 0.000 | −61.3344 | −27.5156 |
| | con_3 | 84.15700 * | 7.76082 | 0.000 | 67.2476 | 101.0664 |
| | con_4 | −43.29400 * | 7.76082 | 0.000 | −60.2034 | −26.3846 |
| | con_5 | −44.83700 * | 7.76082 | 0.000 | −61.7464 | −27.9276 |
| | con_6 | −51.63167 * | 7.76082 | 0.000 | −68.5410 | −34.7223 |

**Table A3.** *Cont.*

| LSD | | Mean Difference (I–J) | Std. Error | Sig. | 95% Confidence Interval | |
| --- | --- | --- | --- | --- | --- | --- |
| Conditions | | | | | Lower Bound | Upper Bound |
| con_3 | con_1 | −128.58200 * | 7.76082 | 0.000 | −145.4914 | −111.6726 |
| | con_2 | −84.15700 * | 7.76082 | 0.000 | −101.0664 | −67.2476 |
| | con_4 | −127.45100 * | 7.76082 | 0.000 | −144.3604 | −110.5416 |
| | con_5 | −128.99400 * | 7.76082 | 0.000 | −145.9034 | −112.0846 |
| | con_6 | −135.78867 * | 7.76082 | 0.000 | −152.6980 | −118.8793 |
| con_4 | con_1 | −1.13100 | 7.76082 | 0.887 | −18.0404 | 15.7784 |
| | con_2 | 43.29400 * | 7.76082 | 0.000 | 26.3846 | 60.2034 |
| | con_3 | 127.45100 * | 7.76082 | 0.000 | 110.5416 | 144.3604 |
| | con_5 | −1.54300 | 7.76082 | 0.846 | −18.4524 | 15.3664 |
| | con_6 | −8.33767 | 7.76082 | 0.304 | −25.2470 | 8.5717 |
| con_5 | con_1 | 0.41200 | 7.76082 | 0.959 | −16.4974 | 17.3214 |
| | con_2 | 44.83700 * | 7.76082 | 0.000 | 27.9276 | 61.7464 |
| | con_3 | 128.99400 * | 7.76082 | 0.000 | 112.0846 | 145.9034 |
| | con_4 | 1.54300 | 7.76082 | 0.846 | −15.3664 | 18.4524 |
| | con_6 | −6.79467 | 7.76082 | 0.398 | −23.7040 | 10.1147 |
| con_6 | con_1 | 7.20667 | 7.76082 | 0.371 | −9.7027 | 24.1160 |
| | con_2 | 51.63167 * | 7.76082 | 0.000 | 34.7223 | 68.5410 |
| | con_3 | 135.78867 * | 7.76082 | 0.000 | 118.8793 | 152.6980 |
| | con_4 | 8.33767 | 7.76082 | 0.304 | −8.5717 | 25.2470 |
| | con_5 | 6.79467 | 7.76082 | 0.398 | −10.1147 | 23.7040 |

* The mean difference is significant at the 0.05 level.

**Table A4.** Dependent Variable: Total dissolved Nitrogen uptake (%).

| LSD | | Mean Difference (I–J) | Std. Error | Sig. | 95% Confidence Interval | |
| --- | --- | --- | --- | --- | --- | --- |
| Conditions | | | | | Lower Bound | Upper Bound |
| con_1 | con_2 | 49.71667 * | 6.95668 | 0.000 | 34.5594 | 64.8740 |
| | con_3 | 129.06000 * | 6.95668 | 0.000 | 113.9027 | 144.2173 |
| | con_4 | 22.91333 * | 6.95668 | 0.006 | 7.7560 | 38.0706 |
| | con_5 | 7.83333 | 6.95668 | 0.282 | −7.3240 | 22.9906 |
| | con_6 | 5.57000 | 6.95668 | 0.439 | −9.5873 | 20.7273 |
| con_2 | con_1 | −49.71667 * | 6.95668 | 0.000 | −64.8740 | −34.5594 |
| | con_3 | 79.34333 * | 6.95668 | 0.000 | 64.1860 | 94.5006 |
| | con_4 | −26.80333 * | 6.95668 | 0.002 | −41.9606 | −11.6460 |
| | con_5 | −41.88333 * | 6.95668 | 0.000 | −57.0406 | −26.7260 |
| | con_6 | −44.14667 * | 6.95668 | 0.000 | −59.3040 | −28.9894 |

**Table A4.** *Cont.*

| LSD | | Mean Difference (I–J) | Std. Error | Sig. | 95% Confidence Interval | |
|---|---|---|---|---|---|---|
| Conditions | | | | | Lower Bound | Upper Bound |
| con_3 | con_1 | −129.06000 * | 6.95668 | 0.000 | −144.2173 | −113.9027 |
| | con_2 | −79.34333 * | 6.95668 | 0.000 | −94.5006 | −64.1860 |
| | con_4 | −106.14667 * | 6.95668 | 0.000 | −121.3040 | −90.9894 |
| | con_5 | −121.22667 * | 6.95668 | 0.000 | −136.3840 | −106.0694 |
| | con_6 | −123.49000 * | 6.95668 | 0.000 | −138.6473 | −108.3327 |
| con_4 | con_1 | −22.91333 * | 6.95668 | 0.006 | −38.0706 | −7.7560 |
| | con_2 | 26.80333 * | 6.95668 | 0.002 | 11.6460 | 41.9606 |
| | con_3 | 106.14667 * | 6.95668 | 0.000 | 90.9894 | 121.3040 |
| | con_5 | −15.08000 | 6.95668 | 0.051 | −30.2373 | 0.0773 |
| | con_6 | −17.34333 * | 6.95668 | 0.028 | −32.5006 | −2.1860 |
| con_5 | con_1 | −7.83333 | 6.95668 | 0.282 | −22.9906 | 7.3240 |
| | con_2 | 41.88333 * | 6.95668 | 0.000 | 26.7260 | 57.0406 |
| | con_3 | 121.22667 * | 6.95668 | 0.000 | 106.0694 | 136.3840 |
| | con_4 | 15.08000 | 6.95668 | 0.051 | −0.0773 | 30.2373 |
| | con_6 | −2.26333 | 6.95668 | 0.751 | −17.4206 | 12.8940 |
| con_6 | con_1 | −5.57000 | 6.95668 | 0.439 | −20.7273 | 9.5873 |
| | con_2 | 44.14667 * | 6.95668 | 0.000 | 28.9894 | 59.3040 |
| | con_3 | 123.49000 * | 6.95668 | 0.000 | 108.3327 | 138.6473 |
| | con_4 | 17.34333 * | 6.95668 | 0.028 | 2.1860 | 32.5006 |
| | con_5 | 2.26333 | 6.95668 | 0.751 | −12.8940 | 17.4206 |

* The mean difference is significant at the 0.05 level.

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
