# Peer review of "Exploration of Microalgae-Activated Sludge Growth Performance in Lab-Scale Photobioreactors under Outdoor Environmental Conditions for Wastewater Biotreatment"

_phycology, doi:10.3390/phycology3040033_

Round 1

Reviewer 1 Report

Comments and Suggestions for Authors

Author Response

Dear Editor and Reviewers,

I am writing on behalf of the co-authors to appreciate your time and efforts in processing our manuscript and evaluating our work publishing in the Phycology.

We appreciate the reviewers’ and editorial board's constructive comments and recommendations, also the opportunity to improve the manuscript. We have carefully considered all comments and have made changes to improve the quality of the manuscript accordingly.

The corrections and comments are described in the attached file.

Kind regards,

Dr. Ruas Graziele

(on behalf of the Authors)

Reviewer 2 Report

Comments and Suggestions for Authors

This study aims to evaluate the growth performance of native microalgal activated sludge (MAS) for the tertiary treatment of anaerobically digested wastewater from an upflow anaerobic sludge blanket (UASB) in laboratory-scale open-air photobioreactors. free under different conditions with different inoculum concentrations operating in batches. The performance of the system is studied from the point of view of biomass growth, nutrient and coliforms removal. This study is presented as innovative because it was carried out in outdoor environmental conditions. There is no extensive literature on tertiary treatment of UASB effluents through microalgae cultivation, so the study provides new valuable information for the scientific community. However, there are some pilot-scale studies that have already been carried out and have not been cited in the bibliography that will be suggested later in the comments with the intention of improving the manuscript.

(Major revision)

First, I present some conceptual and methodological doubts that, in my opinion, should be reviewed in depth.

On the one hand, the author presents 7 different conditions, but in fact they are 3 conditions to be analyzed. The others are actually controls from the previous three, since the study carried out focuses on the performance of a mixotrophic culture of microalgae and activated sludge (scenarios 1, 2 and 3). the scenarios 4, 5 and 6 are in fact the controls of the first three respectively. The last one, scenario 7, is a condition without microalgae inoculum neither activated sludge and it seems no relevant for this study. Since it is only mentioned in line 248 to be compared with the scenarios 2 and 3, the performance of which was inappropriate. It is kindly suggested reframing the manuscript's discourse in this sense. Modify line 18 of abstract "seven conditions" to "three conditions" and refer only to these three conditions studied (1, 2 and 3) and their controls (4, 5 and 6) in whole the manuscript. In case the author considers that condition 7 is relevant to the study, it is necessary to develop more extensively the importance of condition 7 in each subsection of results and discussion.

On the other hand, the experiment setup was realized by mean volumetrically. The different configurations have different concentrations of microalgae and activated sludge. The way to obtain each desires concentration was carry out measuring the concentration in the microalgae and sludge medium and calculating which is the volume necessary to add in the photobioreactor of 2 L to achieve the desired concentration of the configuration. However, the different volumes of microalgae and sludge which causes having a different volume of anaerobically digested wastewater medium in each configuration. That makes to having different concentrations of nutrients and other compounds and therefore, different initial conditions in each configuration. This seriously compromises the results presented in the manuscript, since they are not directly comparable.

The common procedure to avoid this inconvenience is to centrifuge the volume of the biomass that is desired to be obtained (in this case microalgae and activated sludge) to be later added into the medium. In this way, the volumetric problem and the addition of other components that may be found in the source media (microlages and sludge) are avoided. For all this, it is strongly recommended to do a thorough review of the data, recalculating the results obtained and adjusting them proportionally to the volumes of each configuration to know what the real performances have been. The manuscript must even include the information that the possible nutrients and compounds that may have been added in different proportions to the photobioreactors of each configuration, coming from the volumes of the algae and sludge media, were not taken into account.

It is essential that this aspect be addressed correctly, so that the article can be published, since incorrect results can cause confusion to other researchers who take this manuscript as a reference for their research.

Also, in chapters 3.3 and 3.4: In both sections it is mentioned that the best scenario of those studied (1, 2 and 3) is 1, the one with the lowest sludge inoculum. As the author indicates, this is due to being the configuration with the least concentration of sludge and providing less turbidity. This allows a greater amount of light so that the algae can carry out photosynthesis and therefore increase their growth and greater removal of nutrients.

However, there is no scenario with a smaller amount of activated sludge inoculum that corroborates that this is the optimal dose. It could simply be argued that the less inoculum the better for the culture of mycoalgae and that this does not provide any benefits.

If it is not possible to compare this result with a lower concentration to ensure that the inoculum provides a specific benefit with respect to lower concentrations, the discussion of the results should be focused in one sense: "what maximum amount of inoculum is the system capable of accepting?" without compromising performance" and not "what is the optimal solution for the system".

Finally, biomass growth yields in terms of TSS have been studied in the manuscript. However, not always 100% of the TTS are biomass, since there may be a significant percentage of inorganic solids. VSS are usually used to ensure that the result obtained is 100% biomass. If the author can provide additional information that ensures that 100% of the TSS are biomass, it should be included in the manuscript. If the author cannot ensure this, this information should be added to the results section.

(Minor revision)

Following, a review of minor aspects related to the text and some minor concepts is made indicating the lines of text to which are refered.

Line 19: “…in triplicate…”. This coment can be omitted in the abstract. It is an information only proper for the materials and methods chapter.

Lines 60-64: The author do not mentions the others studies in outdoor with the same characteristics. For example, in Lucas Vassalle et al. 2020 “Can high rate algal ponds be used as post-treatment of UASB reactors to remove micropollutants? a UASB - High rate Algal Pond was operated and evaluated at outdoor pilot-scale.

Lines 72-78: The author takes into account the energy savings of exhaustive control in laboratory conditions, but does not mention the loss of efficiency that may occur outdoors due to the variability of environmental conditions.

Lines 144-145: The phrase “The experiment was conducted for 5 days, and thus assumed 5 days operational HRT”. Usually, a star-up conditions, as an experiment in batch, can not be directly compared assuming the same HRT in a reactor operating in continuous conditions. This phrase should be reformulated or eliminated.

Lines 147-148: The make and model of the pump has already been described in the materials and methods section. It is not necessary to describe the pump again and this information could be omitted, simply stating that the correct agitation is thanks to the pump.

Line 231: “…experiment. the same which…” for “…experiment. The same which…”

Lines 322-323: “…Based on the results obtained in nutrients removal, observable increases in nutrients uptake were recorded for conditions 1, 4, 5, 6, and 7,”. The author says that the nutrients removal increases, It is not specified regarding what. It is not specified regarding what. Reference should be made regarding the scenario in which this increase occurs or the text reformulated.

Table 5: The numbers of the different scenarios should be indicated in the table.

Author Response

(The authors gave the same response as above.)

Round 2

Reviewer 2 Report

Comments and Suggestions for Authors

The author has addressed the major issues proposed by the reviewer assertively and addressing all of minor issues. The manuscript is considered suitable for publication and contributes to the scientific community with relevant data in the field of wastewater treatment. Specifically in the treatment based on microalgae cultures and anaerobic digestion. The study has been developed in outdoor conditions, which provides added value and valuable information for future studies of scaling these processes at an industrial level.